# Geographic Medical Overview of Noncommunicable Diseases (Cardiovascular Diseases and Diabetes) in the Territory of the AP Vojvodina (Northern Serbia)

**DOI:** 10.3390/healthcare11010048

**Published:** 2022-12-23

**Authors:** Emina Kričković, Tin Lukić, Dejana Jovanović-Popović

**Affiliations:** 1Faculty of Geography, University of Belgrade, Studentski Square 3/III, 11000 Belgrade, Serbia; 2Department of Geography, Tourism and Hotel Management, Faculty of Sciences, University of Novi Sad, Dositeja Obradovića Square 3, 21000 Novi Sad, Serbia; 3The Faculty of Security Studies, University of Belgrade, 50 Gospodara Vučića Street, 11118 Belgrade, Serbia

**Keywords:** noncommunicable diseases, AP Vojvodina, cardiovascular diseases, diabetes, population health, geographic medical overview

## Abstract

The objective of this study was a geographic medical analysis of noncommunicable diseases (cardiovascular diseases from 2010 to 2020 and diabetes from 2010 to 2019) in the AP Vojvodina (northern Serbia) in order to identify the most and least burdened counties as well as to present trends in the mentioned diseases. The Mann-Kendall trend test, a cluster analysis, and Getis–Ord Gi* method for hot spot analysis were applied in this analysis. Regarding acute coronary syndrome and myocardial infarction, the North Backa County had a lower mortality rate although the number of newly reported cases was above average. The largest number of new cases of unstable angina pectoris was in the North Backa, North Banat, and Middle Banat Counties, while the West Backa County was identified as a county with a higher mortality rate. The cluster analysis showed that the number of death cases from diabetes in the Srem County is significantly higher than that in the other counties. Likewise, the West Backa County had a high number of new diabetes patients, but also a much lower mortality rate. Chronic noncommunicable diseases are predominant in newly diagnosed incidences and death cases in the AP Vojvodina. Studies of this kind promote public health and healthcare systems in the researched area and in the Republic of Serbia, as well as in other countries.

## 1. Introduction

This work represents a part of the study “The effect of geographic medical factors on the health of the population of the AP Vojvodina” which can be regarded as a pioneer research work in the field of the medical geography of Serbia. The Autonomous Province of Vojvodina (AP Vojvodina, northern Serbia) is a geospatial entity with specific natural-geographic and social-economic features and represents the researched area in this work. The aetiology of most chronic noncommunicable diseases (NCDs) is quite complex and it is very difficult to identify, with certainty, the factors which cause the abovementioned diseases. The health of people depends significantly on the conditions present in the environment and is related to its influences. A great number of factors, the most common being smoking, inadequate diet habits, insufficient physical activity, stress, and environmental pollution, influence the occurrence of noncommunicable diseases. A look back at the health status of the population in Vojvodina in the second half of the 20th century reveals that the communicable diseases which used to pose leading health issues in the past are now under control [1]. The abovementioned factors can be attributed to a rapid socio-economic development, vaccination, conduction of programs for combating, eliminating, and eradicating communicable diseases, environmental recovery, and an increase in health culture. It is also necessary to acknowledge the fact that certain demographic changes take place—first and foremost, the ageing of the population, which leads to a higher number of people suffering from mass noncommunicable diseases [1].

According to WHO’s NCD Portal, in 2019, noncommunicable diseases were responsible for 74% of total deaths [2]. Furthermore, 72% of all male deaths and 75% of all female deaths were attributed to noncommunicable diseases [2]. In the same year, the world’s mortality rate of NCDs was 479 per 100,000 population, out of which the male mortality rate was higher than the female one, i.e., 576 to 397 per 100,000 population, respectively [2]. The mortality rate of NCDs in Europe, in the same year, was 427.15 per 100,000 population. According to the WHO report on noncommunicable diseases profiles across countries, the burden is the greatest within low and middle-income countries, where 78% of all noncommunicable disease deaths and 85% of premature deaths occurred [3]. The probability of premature mortality from NCDs in the world in 2019 was 18% and in Europe it was 16% [2]. The mortality rate overview of noncommunicable diseases in Europe in 2019 is displayed in Figure 1, and Serbia is in 15th place among 50 countries.

Analyzing the data from the portal in 2019, one can see that Serbia is above the world average with NCDs responsible for 95% of total deaths (94% of male and 96% of female deaths) [2]. Following this, Serbia is also above the world and European average NCDs mortality rate with over 600 per 100,000 population, together with male and female NCDs mortality rates, i.e., 716 and 503 per 100,000 population, respectively. In addition, the probability of premature mortality from NCDs in Serbia is higher than the world average of 22%. [2] Furthermore, according to WHO’s NCD Data Portal, Serbia had a 600.81 noncommunicable diseases mortality rate in 2019, which is higher than the European average of 427.15 [2].

Cardiovascular diseases pose a grave global problem nowadays. In 2019, there were approximately 12.7 million new cases of cardiovascular diseases (CVDs) in countries with data available in Europe [4]. According to Timmis, A. et al. (2022), in the same year, Serbia had an incidence rate of 920 per 100,000 population and was in 18th place among 54 countries regarding the CVDs incidence rate [4].

Maksimovic et al. (1999) researched the relationship between magnesium and calcium in potable water and CVDs in 65 municipalities in Serbia while Velicki (2017) researched the relationship between diet and acute coronary syndrome among patients with this disease in the territory of the AP Vojvodina [5,6]. Poloniecki et al. (1997) and Jevtic et al. (2014) researched the relationship between air pollution and the occurrence of CVDs in the researched area [7,8].

Diabetes is a disease which represents a major public health issue in today’s world [9]. Over 90% of patients are diagnosed with type 2 diabetes. The highest number of people with the abovementioned diabetes type are between 40 and 59 years of age although more and more younger people have recently been diagnosed with it [9]. Samuelsson, U. and Löfman, O. (2004) within the study “Geographical mapping of type 1 diabetes in children and adolescents in southeast Sweden” established geographical variations of risks of type 2 diabetes incidences in children and adolescents among the municipalities in the region mentioned [10]. Samuelsson, U. et al. (2020) studied geographical variations in the rate of type 1 diabetes incidence in Nordic countries [11]. In the study “Association between Arsenic in Drinking Water and the Occurrence of Type 2 Diabetes”, Jovanovic D. (2013) studied the relationship between arsenic concentrations in potable water in the city of Zrenjanin (AP Vojvodina) and type 2 diabetes cases [12].

According to the International Diabetes Federation’s Diabetes Atlas (IDF Diabetes Atlas), in 2021, Serbia was among the top five countries in Europe with the highest age-adjusted prevalence of people with diabetes (20–79 years)—9.1% [13]. In the same publication, it is stated that 1.1 million deaths in Europe were caused by diabetes in 2021 [13].

In the AP Vojvodina, in 2019, the standardized incidence rate for type 1 diabetes from 0–29 years was 12.7 and for type 2 diabetes it was 157.6 [14]. In the same year, the mortality rates of type 1 and type 2 diabetes in the researched area were 6.0 and 5.9, respectively [14].

The health protection of the AP Vojvodina population is organized within 93 healthcare facilities through the primary, secondary, and tertiary levels [15]. The number of doctors per 1000 population in the province is below the national and the European average—2.48 [16]. In Europe, there are 3.22 and in Serbia, there are 2.86 doctors per 1000 population [16]. In the Republic of Serbia, the registry for coronary heart disease and the registry for diabetes have been legally established since 1980. The population registry for acute coronary syndrome was established in 2006 and the reorganization of the registry for diabetes in Serbia was initiated in the same year [14,17]. These registries are conducted at the regional level while the database is established, maintained, and updated at the national level [14,17].

This work will not examine the causes of noncommunicable diseases occurrence (cardiovascular diseases and diabetes) but will provide a geographic medical overview of the abovementioned diseases in the territory of the AP Vojvodina. This overview will be supported by a hot spot analysis of the mentioned diseases, by the incidence and mortality rates as well as by analyzing the trends of the diseases using well-known methods such as cluster analysis, the Mann–-Kendall test, and emerging hot spot analysis using the Getis–Ord Gi* spatial clustering statistic method.

There are numerous studies describing the geospatial distribution of diseases using emerging hot spot analysis. For instance, Bahri, M.A.S et al. (2014) in the study “Comparison of Spatial Autocorrelation Analysis Methods for Distribution Pattern of Diabetes Type 2 Patients in Iskandar Malaysia Neighbourhoods” used Getis–Ord G* Statistics to compare diabetes type 2 patients’ data both in global and local scales [18]. Furthermore, Kuznetsov and Sadovskaya (2021) and Shariati et al. (2020) used hot spot analysis to understand the spatial distribution of COVID-19 at regional and global scales [19,20]. McEntee and Ogneva Himmelberger (2008) carried out a hot spot analysis for the exposure of diesel particulates using a *t*-test to determine significant differences between the influence of lung cancer and asthma cases [21]. Razavi Termeh, S.V. et al. (2021) in the study “Asthma-prone areas modelling using a machine learning model” used the Getis–Ord Gi* index to map high-risk areas for asthma [22] and Mahara, G. et al. (2018) identified the hot spots of tuberculosis incidence in urban districts using space–time cluster analysis of tuberculosis incidence in Beijing, China [23].

## 2. Materials and Methods

The objective of this work is to research noncommunicable diseases (cardiovascular diseases and diabetes) in the territory of the AP Vojvodina (northern Serbia) in order to identify the most burdened counties and the present trends in the diseases in question.

The data for certain noncommunicable diseases within the research area are not publicly available for a continuous time period. Therefore, the time frame of the research mostly depended on the very accessibility of the available data. In addition, for certain diseases, the records are not published, i.e., the public is not informed due to a legal regulation which enables relevant institutions to decide on the scope and continuity of informing the public. The data on the newly diagnosed and those who have died from cardiovascular diseases were taken from the publication “Incidence and mortality from acute coronary syndrome in Serbia” from 2010 to 2020, and this was issued by the registry of acute coronary syndrome in Serbia and the Institute of Public Health of Serbia “Dr Milan Jovanovic Batut” [17,24,25,26,27,28,29,30,31,32,33]. Likewise, we used “Health statistical yearbooks of the Republic of Serbia” from 2010 to 2020 and these were published by the Institute of Public Health of Serbia “Dr Milan Jovanovic Batut” [16,34,35,36,37,38,39,40,41,42,43], as well as the publications “Health condition of the population of the AP Vojvodina” released by the Institute of Public Health of Vojvodina [15,44,45,46,47,48,49,50,51,52,53,54]. The data on the number of new patients and the deceased as well as on the incidence and mortality rates from diabetes were obtained from the publications “Diabetes incidence and mortality” from 2010 to 2019 [14,55,56,57,58,59,60,61,62,63] published by the Diabetes Registry of Serbia and the Institute of Public Health of Serbia “Dr Milan Jovanovic Batut”.

The researched area represents a connection between central and western Europe on one side and the Near East and the Balkans on the other. The Autonomous Province of Vojvodina is located in the north of the Republic of Serbia and boasts a favorable geographical position [64]. It is located between 44°38′ and 46°10′ north latitude and between 18°10′ and 21°15′ east longitude, in south-eastern Europe, in the Balkans, and comprises the southern part of the Pannonian Basin [64].

To the north, it is bordered by Hungary, to the east by Romania, to the west by Croatia, whereas to the south-east it is bordered by Bosnia and Herzegovina, and to the south by the Danube and Sava rivers [65]. The researched area covers the surface of 21.506 km^2^. Bearing in mind the fact that as many as ten European countries occupy a smaller territory (Slovenia, Montenegro, Cyprus, Luxembourg, Andorra, Malta, Liechtenstein, San Marino, Monaco, and the Vatican) and observing the area of Vojvodina, one can conclude that it is a relatively large entity of regional autonomy [66].

The researched area accounts for 24.3% of the entire surface of the Republic of Serbia and it has the population of 1.93 million, which constitutes 27.2% of the entire population number if we do not take into account the territory of the AP Kosovo and Metohija [66].

In this research the following methods were applied: cluster analysis, the Mann–-Kendall test, and the cartographic method. For the sake of mapping, the data on the diseases were used at the county level, whereas for the analysis via the Mann–Kendall test, the data were used on the level of the province. The incidence and mortality rates were used for analyzing a disease. The cluster analysis was applied to identify the most burdened counties and also for the areas in which the mentioned diseases are less present. The cluster analysis represents a statistical method of processing the data by means of a number of operations over the data in order to obtain, transform, or classify the information within the data set being monitored. The cluster analysis functions by the principle of organizing the data into groups or clusters in relation to how these data are related, i.e., grouping the data or the objects in such a way that the data or objects within the same group are more similar to each other than to the ones in the other groups/clusters. The cluster analysis serves more as a research tool than a means of prediction.

With the aim of statistical examination, we will use the k-center method which is mostly used in the statistical cluster analysis. This algorithm attempts to find groups by minimizing the distance between the data.

The calculation of the distance between the data is Euclidean, and it is performed by means of the formula below:(1)$$d\left(x,y\right)=\sqrt{\overset{2}{\underset{i=1}{\sum }}{\left(x_{i}-y_{i}\right)}^{2}}$$
in which *x* = (*x*_1_,*x*_2_) = (newly diagnosed or dead) is a coordinate point from the chart, whereas *y* is a point with different coordinates *y* = (*y*_1_,*y*_2_ ).

During the cluster analysis, the statistical program R Studio was used; it contains the function k means for calculating the k-center and grouping the data for a given number of clusters. It is quite significant to point out here that before we apply the k-center (k means) algorithm, we need to assign a number of clusters. In case we are not sure about the number of clusters, we can apply the following formula to calculate it:(2)$$No.\;of\;clusters=\frac{\sqrt{n}}{2}$$
where *n* stands for the total number of the data obtained.

As a trend test, this study applies the widely used Mann–-Kendall test, a non-parametric test assuming stable, independent, and random time series with equal probability distributions [67]. In this research, it will be used in order to examine the trends in the most frequent noncommunicable diseases in the studied area where we will also examine which diseases tend to increase/decline. By means of the abovementioned method, one can estimate how a variable can change when influenced by another one. The MK test is simple and robust; it can cope with missing values and values below the detection limit [68]. Starting from the first test suggestions by Mann– [69] and Kendall [70], the test itself was later extended for a wider application [71]. The Mann–-Kendall test can be applied in cases when it can be assumed that for the value of xi time series, the following model holds true [68]:(3)$$S=\overset{n}{\underset{i=2}{\sum }}\overset{i-1}{\underset{j=1}{\sum }}sign\left(x_{i}-x_{j}\right)$$
where sign $x_{i}-x_{j}$ is:(4)$$sign\left(x\right)=\{\begin{array}{c} 1,\;\;for\;\;x_{i}-x_{j}>0\\ 0,\;\;for\;\;x_{i}-x_{j}=0\\ -1,\;\;for\;\;x_{i}-x_{j}<0 \end{array}\;$$

The statistic *S* tends to normality for a large n, with a mean and a variance defined as follows [69],
(5)E(S)=0
(6)$$V\left(S\right)=1/18\left[n\left(n-1\right)\left(2n+5\right)-\overset{q}{\underset{P=1}{\sum }}t_{P}\left(t_{P}-1\right)\left(2t_{P}+5\right)\right]$$
where *n* is the length of the times-series, $t_{P}$ is the number of ties for the $p^{th}$ value, and *q* is the number of tied values (i.e., equals values). The second term represents an adjustment for tied or censored data. The standardized test statistic Z is given by [72,73]:(7)$$Z=\{\begin{array}{cc} \frac{S-1}{\sqrt{Var\left(S\right)}} & if\;S>0,\\ 0 & if\;S=0,\\ \frac{S+1}{\sqrt{Var\left(S\right)}} & if\;S<0, \end{array}\;$$

The presence of a statistically significant trend is evaluated using the *Z* value. This statistic is used to test the null hypothesis such that no trend exists [72]. A positive *Z* indicates an increasing trend in the time-series, while a negative *Z* indicates a decreasing trend [72]. To test either an increasing or decreasing monotonic trend at the *p* significance level, the null hypothesis is rejected if the absolute value of *Z* is greater than $Z_{1-P/2}$; where $Z_{1-P/2}$; is obtained from the standard normal cumulative distribution tables [72]. According to the MK test, two hypotheses were tested: the null hypothesis, H_0_, stating that there is no trend in the time series; and the alternative hypothesis, Ha, stating that there is a significant trend in the series for a given significance level limit [68]. The probability, p, was calculated to determine the level of confidence in the hypothesis. If the computed *p*-value is lower than the chosen significance level α (e.g., α = 5%), the H_0_ (there is no trend) should be rejected and the H_a_ (there is a significant trend) should be accepted; and if the *p*-value is greater than the significance level α, then the H_0_ is accepted (or cannot be rejected) [74]. For calculating the probability (p) and for hypothesis testing, XLSTAT statistical analysis software was employed [74].

The cartographic method was applied to examine the geographical distribution of cardiovascular diseases and diabetes within the researched area. ArcGIS Pro software (a product of the American ESRI company) was used. 

We used the Getis–Ord $G_{i}^{*}$ statistic within the emerging hot spot analysis (Space Time Pattern Mining) tool to determine the areas characterized as “hot spots and cold spots” with regard to noncommunicable diseases (cardiovascular diseases and diabetes). This classifies the spatial patterns into clusters and outliers where the former can either be positive (hot spots) or negative (cold spots), while the latter are spatial objects whose attribute values are distinctly different from those of their spatial neighbors [75]. The space–time analysis we used in ArcGIS Pro uses a space–time cube (netCDF cube) as an input and identifies trends in data, such as new, intensifying, diminishing, and sporadic hot and cold spots. The space–time cube in ArcGIS Pro was made of defined locations (counties) and it structured time-stamped features into a netCDF data cube by generating space–time bins with defined features (in this case counties) with the associated spatiotemporal attributes. By creating this cube, the statistic values were calculated and the trend for bin values across time in each county was measured using the Mann–-Kendall statistical test.

This hot spot analysis utilizes the $G_{i}^{*}$ statistic that can be calculated as follows:(8)$$G_{i}^{*}=\frac{\sum_{j=1}^{n}\omega_{i,j}x_{j}-\bar{X}\;\sum_{j=1}^{n}\omega_{i,j}}{S\sqrt{\frac{n\sum_{j=1}^{n}\omega_{i,j}^{2}-{\left(\sum_{j=1}^{n}\omega_{i,j}\right)}^{2}}{n-1}}}$$
where $G_{i}^{*}$ is the spatial autocorrelation (spatial dependency) statistics of an event *i* over the *n* events, the term $x_{j}$ defines the magnitude of the variable *x* at the events *j* over all *n*, and the term $\omega_{i,j}$ defines the weight value between the events *i* and *j* that represents their spatial interrelationship [76,77,78] and:(9)$$\bar{X}=\frac{\sum_{j=1}^{n}x_{j}}{n}$$
(10)$$S=\sqrt{\frac{\sum_{j=1}^{n}x_{j}^{2}}{n}}-{\left(\bar{X}\right)}^{2}$$

The $G_{i}^{*}$ statistics consider the magnitude of each feature in the data set in the context of its neighbours’ values [76,77,78]. The local sum of a feature and its neighbors has been compared to the sum of all features. If there is a significant difference between the local sum and the expected local sum, where the difference is too large due to randomness, a statistically significant z-score is the result [76,77,78]. With the resultant trend z-score and the *p*-value for each location with data, and with the hot spot z-score and the *p*-value for each bin, the emerging hot spot analysis tool classification follows the official ESRI’s ArcGIS Pro 3.0 categorization scheme (pattern name and definition) for each study area location.

The utilization of geographic information systems (GIS) for spatial representation of diseases and the conditions in the health sector (government, provinces, and municipalities) of the given population can contribute to its intensive development and applicative importance according to Kričković et al. (2022) [79]. For this purpose, ArcMap software was applied in this study.

Figure 2 presents the integration of the abovementioned methods with the aim of analyzing noncommunicable diseases (cardiovascular diseases and diabetes) within the researched area.

## 3. Results

According to the publications by the Institute of Public Health of Vojvodina [15,44,45,46,47,48,49,50,51,52,53,54], among the noncommunicable diseases in the researched area, the most significant public health problems are blood circulatory system diseases and diabetes. As it has been defined by the methodology of the research, in this part of the work we will present the results of the cluster analysis and the cartographic overviews made in the already mentioned ArcMap software, along with the trends in the most common noncommunicable diseases in the researched area. 

### 3.1. The Results of the Cardiovascular Diseases Research

In 2019, blood circulatory system diseases accounted for 50.5% of all the death causes in the population of the researched area while in 2005 the percentage was 56.8% [15,44]. Table A1 presents the data on the number of new patients and the ones who died from acute coronary syndrome, as well as the incidence and mortality rates from 2010 to 2020.

In order to compare the data on the number of the newly diagnosed, the dead, and the incidence and mortality rates in the course of different years within the analyzed period, we will provide an overview for the following three years (2020, 2015, and 2010). During 2020 the total number of new patients with acute coronary syndrome was 4421, whereas the incidence rate at the level of the entire AP Vojvodina territory was 240 [17]. The highest incidence rate was registered in the North Backa County, whereas the lowest rate was in the South Backa County. The mortality rate in the aforementioned year equaled 52.9 for the entire researched area. The highest mortality rate in 2020 was recorded in the North Banat County, and the lowest rate was recorded in the North Backa County [43]. In 2015, 5502 new patients with the disease in question were registered; more men than women (3448 men and 2054 women) were registered [51]. The incidence rate of 290.8 per population of 100,000 was higher in men than in women. There were 1206 death events (738 men and 468 women died) [51]. During 2010, the number of new cases of acute coronary syndrome was registered as 3399 men and 2195 women [24]. The highest incidence rate was recorded in the area of the North Banat County, whereas the lowest rate was in the North Backa County. The mortality rate in 2010 reached the highest value in the Middle Banat County, and the lowest rate was in the Srem County [46]. Figure 3 presents the ratio of the number of deceased (mortality rate) and that of new patients (incidence rate) with acute coronary syndrome during the period from 2010 to 2020. Each dot in this chart represents the ratio between the incidence and mortality rate for each year in the observed period. A chart was used to determine if there was a simple linear regression in the data over time. As it was determined there is no linear regression and the cluster analysis, the Mann–-Kendall test, as well as the hot spot analysis were applied to determine the disease trends and hot spots.

By means of the cluster analysis and by applying the k-means function in the R programming environment, we have noticed that one cluster particularly stands out—the North Backa County (Figure 4). Unlike the situation in other counties, this county has a much lower mortality rate from acute coronary syndrome although it has an average or even higher than average number of new patients.

However, if we opt to observe two instead of three clusters, we will come up with Figure 5. In that Figure, there are two clusters presented, each of them includes a few different counties. The Middle Banat and North Banat counties diverge from the others (red color in cluster 1) and if we look at the table data overview, we will notice that these two counties have above average numbers of new patients and deceased ones from acute coronary syndrome per population of 100,000.

The identified patterns of acute coronary syndrome in the hot spot analysis of the researched area are presented in Figure 6. Regarding the acute coronary syndrome incidence rate per 100,000 population, only in the North Backa County has no trend been identified, while in all of the six other counties, a rising trend has been identified, e.g., “oscillating hot spot”. As for the mortality rate, a rising trend has been identified only in the South Backa and Middle Banat Counties while no trend has been identified in all of the five other counties.

Figure 14a shows the geographical distribution of the acute coronary syndrome incidence, i.e., the average values during the period analyzed. The North Banat County stands out in particular as it has the highest incidence rate, with the Middle Banat and North Backa Counties follow. The lowest incidence rates have been recorded in the South Backa and Srem Counties. Figure 15a presents the mean values of the mortality rate of acute coronary syndrome from 2010 to 2020. The highest average mortality rate was registered in the North Banat and Middle Banat Counties, followed by the West Backa County, whereas the counties with the lowest average mortality rate are North Backa and South Banat.

Apart from acute coronary syndrome, out of the other cardiovascular diseases, we also analyzed myocardial infarction and unstable angina pectoris. During 2020, there were 3848 newly diagnosed patients with myocardial infarction and the incidence rate equaled 209 [17]. The highest incidence rate of the disease in question was recorded in the North Backa County, and the lowest rate was recorded in the Middle Banat County. In the North Banat County, during the same year, the highest mortality rate was registered, whereas the lowest rate was in the West Backa County. 

The number of newly diagnosed patients with myocardial infarction during 2015 was 4669 and the incidence rate was 246.8 [51]. In the Middle Banat County, there was the highest registered rate of newly diagnosed cases in the same year, equaling 305.4. The mortality rate from myocardial infarction in 2015 was 63.1; a total of 1186 patients died [51]. 

In 2010, myocardial infarction was registered in 2944 men and 1829 women [46]. The incidence rate in the territory of the AP Vojvodina was 243.8, whereas the highest incidence rate during 2010 was registered in the Middle Banat County (338.7) [24]. The mortality rate from myocardial infarction in the researched area in 2010 equaled 86.5, whereas the highest mortality rate was recorded in the Middle Banat County (134.5) [24]. Table A2 presents the number of new patients and deceased ones, as well as the incidence and mortality rates of myocardial infarction and unstable angina pectoris during the period from 2010 to 2020.

Figure 7 presents the ratio between the incidence and mortality rates from myocardial infarction during the analyzed time period. As in the previous acute coronary syndrome ratio chart, dots in this chart also represent the ratio between the incidence and mortality rates for each county. It can be noticed that there is certain linear regression, but it is also visible that the data are dispersed. It could be concluded that the greater the number of newly diagnosed, the t deceased from myocardial infarction. As in the previous case, the same methods were used in order to determine disease trends and hot spots.

Another cluster analysis was applied to the data on the number of newly diagnosed patients and those deceased from myocardial infarction in the AP Vojvodina in the period from 2010 to 2020, i.e., the incidence and mortality rates. The issue with this analysis is that we need to estimate beforehand how many clusters we might have judging by the previous chart. The k-center algorithm is not suitable for variables which come in the form of text as it is necessary to calculate the distance between the data. If we happen to not be sure about the number of clusters, according to the mentioned formula (2) in this set of data *n* = 77, the number of clusters that we want to examine equals √77/2 = 4.39 ≈ 4.

Figure 8 presents the identified clusters for myocardial infarction during the analyzed time period. One cluster stands out among the others, and that is the North Backa County where we may clearly see that, although there are between 150 and 250 new patients per 100,000 population, the number of deceased is much lower in relation to other districts in the AP Vojvodina.

The identified patterns in the hot spot analysis of myocardial infarction are given in Figure 9. In this figure, regarding the myocardial infarction incidence rate, only the Srem County has not shown any trend. In four counties—West Backa, North, Middle, and South Banat Counties—“new cold spots” were identified (decreasing trend), and in the North and South Backa Counties, “oscillating cold spots” were identified (decreasing trend). In the same analysis, only the North and South Backa Counties saw a rising trend in the myocardial infarction mortality rate identified as “oscillating hot spots”, while in the other five counties, there was no trend detected.

Figure 14b presents the incidence rates of myocardial infarction in the period from 2010 to 2020. The South Banat County had the highest incidence rate, and the North Banat and Middle Banat Counties followed. The counties with the lowest incidence rate of myocardial infarction are North Backa and Srem. The mortality rates of myocardial infarction in the 2010–2020 period are shown in Figure 15b. The highest mortality rate during the analyzed period was recorded in the North Banat County, and the Middle Banat County comes next. The counties with the lowest mortality rates during the analyzed period were the North Backa County and the South Banat County.

During 2020, the highest incidence rate of unstable angina pectoris was registered in the West Backa County, whereas the lowest rate was in the South Backa County. Overall, 775 newly diagnosed patients were registered, and the incidence rate was 42.1 [17]. The total of 23 patients died in the researched area and the mortality rate in the mentioned year equaled 1.2 [17]. The highest mortality rate was registered in the West Backa County and the lowest rate was registered in the Middle Banat County. In 2015, unstable angina pectoris was confirmed in 833 people and the incidence rate was 44. The total number of deceased patients from the disease in question in 2015 was 20 and the mortality rate equaled 1.1 [29]. The highest incidence rate was 126.6 in the North Backa County and the lowest was registered in Srem County (20.7) [21]. During 2010, unstable angina pectoris was registered in 455 male patients and 366 females [46]. The incidence rate was 41.9, whereas the mortality rate was 2.6. The total of 50 patients died from this disease in the course of 2010. In the South Banat County, there was the lowest incidence rate of unstable angina pectoris in 2010 (21.1), while the highest rate was in the North Banat County (131.6) [46]. The West Backa County had the highest mortality rate during the analyzed year and it equaled 7.8. [46] Figure 10 presents the ratio of the number of deceased and newly diagnosed patients with unstable angina pectoris during the analyzed time period. As in the previous cases, the ratio chart between the incidence and mortality rates was made in order to determine if there is linear regression in the data. Each dot in this chart represents this ratio in each county over the period. In this case, it is evident that there is no linear regression, and the same methods were used as before in order to determine disease trends and hot spots.

The application of the k-means function for the cluster analysis shows that there are three clusters (Figure 11).

The first cluster includes the North Backa, North Banat, and Middle Banat Counties with the highest number of new patients, but also the average number of patients deceased from unstable angina pectoris. The third cluster includes the West Backa County only with a noticeably higher number of patients deceased from unstable angina pectoris. If we also study the next figure (Figure 12) where the clusters are divided according to the number of the deceased, we obtain the result in which the West Backa County clearly diverges from the other counties, provided that we take into account only the number of those deceased.

The identified patterns in the hot spot analysis of unstable angina pectoris for the incidence/mortality rates per population of 100,000 are presented in Figure 13. In this figure, the Srem County has not displayed a trend in the incidence rate, while the Middle Banat County was identified as a “new cold spot” (decreasing trend) and all of the other five counties—West, North, and South Backa, and North and South Banat were identified as “oscillating cold spots” (decreasing trend). North and South Backa and Middle and South Banat have not shown a trend in the mortality rate, while the West Backa and North Banat Counties were identified as “oscillating cold spots” (decreasing trend) and the Srem County was identified as an “oscillating hot spot” (increasing trend).

Figure 14c presents the incidence rates of unstable angina pectoris for the time period from 2010 to 2020. The highest incidence rates during the analyzed period were recorded in the North Backa and North Banat Counties, whereas the lowest ones were in the South Backa, Srem, and South Banat Counties. During the examined time period, the highest unstable angina pectoris mortality rate was registered in the West Backa County and the lowest rate was recorded in the Middle Banat and South Banat Counties (Figure 15c).

### 3.2. The Results of the Diabetes Research in the Studied Area

Table A3 presents the data on the number of the newly diagnosed individuals with type 1 and type 2 diabetes, the number of the deceased, as well as the incidence and mortality rates from 2010 to 2019. Based on the analysis of these data, in 2019, the number of new, type 1 and 2, diabetes patients was 70 (aged 0 to 29), and the incidence rate was 12.2 [14]. There were 5363 new type 2 diabetes patients registered and the incidence rate was 289.6 [14]. According to “The health statistical yearbook of the Republic of Serbia in 2019”, in the researched area, 681 people died and the mortality rate was 32.8 [42].

According to the report “Health of the population of the AP Vojvodina, 2015”, in 2015, 69 new patients with type 1 diabetes were registered and the incidence rate equaled 11.4 (13.3 at women, and 9.6 at men) [51]. In the same period, there were 5563 newly diagnosed patients with type 2 diabetes. The incidence rate of type 2 diabetes was 283.5 (284.5 for men and 282.6 for women) during the same year [51]. According to the data by the Statistical Office of the Republic of Serbia, in 2015 there were 935 deceased from diabetes (339 people from type 1, 416 people from type 2, and 180 people from unspecified type) [51]. In 2015, the mortality rate of type 1 diabetes was 17.9 (16.8 for men and 19.0 for women), whereas the rate of type 2 was 22.0 (19.3 for men and 24.5 for women) [51].

In 2011, the number of new patients registered with type 1 diabetes was 92, out of which 48 were women and 44 were men [56]. In the same period, there was a significantly higher number of newly registered patients with type 2 diabetes. The incidence rate of type 1 diabetes was higher in female patients (14.7) than in male patients (12.7). The incidence rate of type 2 diabetes was 246.9 for men and 247.2 for women. During 2011, 339 people died from type 1 diabetes in the AP Vojvodina as well as 361 people dying from type 2 [47]. The mortality rate of type 1 diabetes was 14.8 for men and 19.9 for women, whereas the mortality rate of type 2 diabetes was 15.6 for men and 21.3 for women [47].

Figure 16 represents the ratio between the diabetes incidence and mortality rates during the analysed period. As in the previous cases, dots represent the ratio between the incidence and mortality rates in each county for each year in the period. It is clearly visible that there is no linear regression and, therefore, the same methods were used in order to determine the disease trends and hot spots as before. For the analysis of these data, the k-center algorithm was also applied, but number three was taken as the number of clusters. The following chart was obtained (Figure 17).

The first cluster clearly includes the values for the Srem County, and it obviously deviates from all the remaining counties. If we take a look back at the table with the data, we can notice that the number of the deceased from diabetes in the Srem County is significantly higher than that in any other counties, although the number of new patients in that county is not significantly higher than the number of new patients in other counties. The second cluster comprises a few counties: the West Backa and Middle Banat Counties. The West Backa County can particularly be set apart as, unlike the Srem County, it may have a high number of newly diagnosed individuals with diabetes (types 1 and 2), but it also has significantly fewer deceased patients. 

Figure 18 represents identified patterns in the emerging hot spot analysis for the debates incidence and mortality per 100,000 population in the researched area. In this analysis, the West Backa and South Banat Counties have not demonstrated any trend in the diabetes type 1 incidence rate, while the North Backa County was identified as a “new cold spot” (decreasing trend) and the North Banat, South Backa, and Srem Counties were identified as “oscillating cold spots” (decreasing trend). Apart from this, the Middle Banat County was identified as an “oscillating hot spot” (increasing trend). Following this, the North Banat and Srem Counties have not shown any trend in the diabetes type 2 incidence rate, while all of the five other counties—West, North, and South Backa and Middle and South Banat—were identified as “oscillating hot spots” (increasing trend). The North Backa and Banat counties have not demonstrated any trend in the diabetes mortality rate, while all of the five other counties—West and South Backa, Middle and South Banat, and Srem were identified as “oscillating hot spots” (increasing trend).

From 2010 to 2019, the highest values of the incidence rates of both diabetes types were recorded in the Middle Banat County, followed by the West Backa, North Banat, and South Banat Counties (Figure 19a). The lowest value of the incidence rate in the course of the analyzed period was registered in the Srem County.

Figure 19b clearly makes it visible that the highest value of the incidence rate of type 1 diabetes was recorded in the North Backa County, followed by North Banat, while the county with the lowest incidence rate of type 1 diabetes was South Banat.

The incidence rate of type 2 diabetes during the studied period, from 2010 to 2020, is also presented in Figure 19c. The highest incidence rate of type 2 diabetes was recorded in Middle Banat, preceding the West Backa, North Banat, and South Banat Counties, while Srem had the lowest value. The highest mortality rate of both diabetes types during the analyzed period was registered in Srem, and it was followed by the North Banat and South Banat Counties. The lowest mortality rate was recorded in the West Backa County, as can be clearly seen in Figure 19d.

### 3.3. The Results of the Trends in Cardiovascular Diseases and Diabetes in the Researched Area

By means of the Mann–-Kendall test, we calculated the trends in newly diagnosed patients and those who are deceased from cardiovascular diseases and diabetes. Table 1 presents the parameters of the Mann–-Kendall test for the abovementioned diseases where one can clearly see the statistical reliability of each test.

From Table 1 and Figure 20a, it can be concluded that, during the analyzed period, the number of new patients with acute coronary syndrome declined, with the probability of the model being 95%. Unlike the number of new patients, with the probability of 99.9% one can claim that the number of the deceased from the disease in question is on the decrease, as shown in Figure 20b. Figure 20c shows the number of the newly diagnosed individuals with myocardial infarction with the overview of the number trend of new patients. The number of the newly diagnosed individuals with the disease in question is on the decrease, but with the test probability of 90%. Analyzing the aforementioned table along with Figure 20d, one can claim with a high probability (99%) that the number of patients deceased from myocardial infarction in the territory of the AP Vojvodina declined during the assigned period. The number of new patients with unstable angina pectoris along with the trend in the number of new patients during the period from 2010 to 2020 is shown in Figure 20e. It can be noted that the number of new patients with the disease in question is on the decrease, but not with such a high reliability and probability, as can be observed in Table 1. Figure 20f presents the trend in numbers of the deceased from unstable angina pectoris in the period from 2010 to 2020. It can be noted that the number of people who died from the abovementioned disease has a decreasing trend, but without any statistical significance, as seen from Table 1.

Analyzing the number of new patients and the number of the deceased from diabetes in the researched area in the period from 2010 to 2019, one can say that the results seem interesting, but without a high reliability of the test applied. Namely, if we observe the number of new type 1 diabetes patients in the AP Vojvodina, then according to Table 1 and Figure 21c, we can claim with a probability of 95% that the number of new patients is decreasing. According to the same table, one can also claim the number of new type 2 diabetes patients is on the rise. The trend growth of the number of new patients with both diabetes types in the AP Vojvodina equals 73.5, whereas its value is 77.3 for the number of new patients with type 2 diabetes. This case with different diabetes types can be explained in the following way: the number of new type 1 diabetes patients does not account for a big percentage in the total number of new patients with both diabetes types, i.e., the number of new type 1 diabetes patients is lower than the number of new type 2 diabetes patients. The number of the deceased is taken as an overall sum for the entire researched area for the abovementioned reason and, in accordance with Figure 21b, we can notice that this number is declining. However, according with Table 1, the previously mentioned test did not prove to be statistically significant.

## 4. Discussion

This study has not dealt with the causes of the occurrence of noncommunicable diseases (cardiovascular diseases and diabetes) but rather with identifying the most and least burdened areas. Despite the fact that cluster analyses rarely prove fruitful in identifying causation, they may—like single case reports—have the potential to generate new knowledge [80]. Growing public awareness of environmental hazards has led to an increased demand for public health authorities to investigate geographical clustering of diseases [80]. Although such a cluster analysis is nearly always ineffective in identifying the causes of a disease, it often has to be used to address public concern about environmental hazards [80]. With research studies of this kind, it is possible to research the areas where certain diseases occur more intensively and thus identify their hot spots. By doing so, we can facilitate carrying out additional spatial analyses in order to examine the causation of certain diseases in different geographical environments.

In this research, hot spot analysis was used to identify more reliable trends. There were certain discrepancies between the results of the classic cluster analysis and the hot spot analysis, which was expected due to a well-known fact that a hot spot analysis provides better results than a classic cluster analysis. The hot spot analysis is much more detailed, and the results differ from this. It is important to note that certain limitations of this procedure can be identified. Due to data scarcity (the data sets for this research were only available for seven counties the AP Vojvodina ), the results obtained by the emerging hot spot analysis (spatial pattern mining) tool are somewhat less significant due to the available data resolution [81]. Due to this, it is not possible to identify hot spots with great statistical significance in contrast to the performed cluster analysis.

The North Backa County is the one with the lowest cardiovascular diseases mortality rate. This represents a positive instance of the approach to health, where we may see that although there is the highest number of new patients with cardiovascular diseases, the number of deceased is the lowest one in the province. Similarly, the fact that the West Backa County was identified as the one with a higher number of people who died from unstable angina pectoris and with a higher number of new patients with diabetes speaks volumes on how serious the issue of the patients’ approach to health is in this county. It is possible to notice that, in this case, the classic cluster analysis and the hot spot analysis showed the same results, with the only difference being within the North Backa County, where the classic cluster analysis showed a decreased mortality rate, and the hot spot analysis did not identify any trends. The Srem County has a significantly higher number of individuals deceased from diabetes than any other county although its number of new patients is not significantly higher than in other counties. Some of possible reasons could be the habit of the Srem County population in not testing their sugar blood level and neither taking precautions nor following therapeutic procedures to avoid certain medical complications and mortality from diabetes.

According to the Demographic Statistics 2018 publication, issued by the Statistical Office of the Republic of Serbia, the age structure of the researched area has changed significantly since 1950 [82]. The median age in 1950 was 30.96 years and in 2018 it was 42.8 [82]. According to the Statistical Office of the Republic of Serbia online database in 2021, the median age in the researched area was 43.1 years and in the entire Republic of Serbia it was 43.5 [83], which are both below the European average of 44.1 years, according to the Eurostat website [84]. Only two counties were below the national average—the South Backa County (41.5 years) and the North Backa County (43.3). In one county, the average was the same as the national median age—in the South Banat County (43.5 years). All of the four other counties had a median age above the national average—the West Backa County (44.9), the North Banat County (44.1), the Middle Banat County (43.8), and the Srem County (43.7) [83]. 

In 2019, life expectancy in Vojvodina for women was 77.9 years and for men it was 72.0 years. Every fifth person in Vojvodina is aged 65 or above (19.7%) [15]. The researched area population belongs to the regressive type of the population, which is defined by a high proportion of the older population and the small participation of young people, considering that 41% of the population is aged 50 and only 14.4% of the population is below 15 [15]. 

In 2020, the highest number of new patients diagnosed with cardiovascular diseases was recorded in the population aged 75 and above—1033 patients with myocardial infarction with a 749.0 incidence rate per 100,000 population, 188 patients with unstable angina pectoris with a 136.3 incidence rate, and 1169 patients with acute coronary syndrome with an 847.6 incidence rate [17]. The highest number of deceased patients with cardiovascular diseases was also recorded in the population category aged 75 and over in the same year, with 387 deceased from myocardial infarction and a 280.6 mortality rate per 100,000 population, with 10 deceased from unstable angina pectoris and a 7.3 incidence rate, and with 397 deceased from the acute coronary syndrome and a 287.9 incidence rate [17].

In 2019, the highest number of patients newly diagnosed with type 2 diabetes was recorded in the population aged from 65 to 69 (995 patients) and in the same category, the incidence rate was 762.8 [14]. In the same year, the diabetes mortality rate was 138.4, when 382 patient deaths were recorded in patients aged 75 and above [14].

The study carried out by Maksimovic et al. (1999) showed that in the territory of the AP Vojvodina, potable waters have a relatively low magnesium content and a medium calcium content [5]. However, the sodium content in these waters is rather high, so further examination of this subject matter is needed [5,85]. The research studies carried out by Poloniecki et al. (1997) and Jevtic et al. (2014) reached the following conclusions: the average daily concentration of NO_2_ is connected with the hospital admission of patients with cardiovascular diseases in the territory of Novi Sad [7,8]. The study carried out by the Institute of Public Health of Vojvodina, which analyzed in 2017 the amount of salt in meals prepared for preschool children, indicated that the amount of salt in all of the three meals provided daily mostly exceeded the recommended daily intake [53]. Velicki (2017) carried out a study on a sample of patients from the researched area who were diagnosed with acute coronary syndrome and the results showed that consumption of certain food types (fruit, vegetables, poultry meat, and olive oil) can contribute to reducing the risk of the mentioned disease [6]. Likewise, consumption of red meat and processed meat food increases the risk of acute coronary syndrome occurrence, which was also confirmed by the aforementioned study [6]. Vorgucin et al. (2011) examined metabolic syndrome in the AP Vojvodina and they came to the conclusion that obesity in the researched area is correlated with metabolic syndrome occurrence [86], which poses a huge risk of diabetes and cardiovascular diseases. Similarly, Radic (2016) in her research drew the conclusion that every fourth adult in Vojvodina is obese and every third person is pre-obese, which also presents a high risk factor for the occurrence of the aforementioned diseases [87]. The study by the title of “Research of the health of the population in Serbia in 2019” indicated that, in the researched area, there was a significantly higher percentage of obese people (25.4%) than in the overall territory of the Republic of Serbia, as people there use animal fat and salty food more often and consume tobacco more [88]. 

Therefore, it is clear that the physical-geographical environment as well as the lifestyle and approach to health can have an immense effect on the occurrence of the mentioned diseases. It is thus necessary to conduct more thorough research in the burdened counties in order to identify the risk factors from the environment and to determine the reasons for the origins of the diseases in question. 

This study should represent the basis for future research on the territory of the AP Vojvodina in order to assess the findings from the aforementioned studies on the causative agents of noncommunicable diseases in the affected areas. By identifying the hot spots and causations, we would reduce the number of new patients and death from noncommunicable diseases. As a result, the costs involved in treating these patients could be reduced. A large number of patients allocate significant financial resources for diagnostics, treatments, and therapeutic procedures so finances are quite often a limiting factor. Patients, in practice, quite frequently neither go to regular checks nor do they go for diagnostic tests due to a lack of money; therefore, these diseases are diagnosed too late when a lot of side complications have also appeared. It is well known that in practice, diabetes often occurs in patients suffering from cardiovascular diseases and the other way round, which indicates the importance of prevention. Mortality from cardiovascular diseases and diabetes can be considerably slowed down and reduced thanks to prevention measures, therapy administration, and carrying out regular medical checks. Consequently, we should have a bigger influence on taking prevention measures and protection from noncommunicable diseases which would particularly be directed towards burdened categories.

As the results of the trend analyses have shown, the number of new patients and the number of deceased from cardiovascular diseases is on the decrease, but when it comes to the number of people who died from unstable angina pectoris, the test did not exhibit statistical significance. In addition, the number of people who died from diabetes is declining, but the test showed no statistical significance. It is notable that the numbers of new patients and those who died from cardiovascular diseases have been significantly lower since the outbreak of the COVID-19 pandemic. This can be explained by an assumption that a certain number of the deceased from noncommunicable diseases were categorized as those who died from COVID-19, since these diseases also contributed to their death. For all of the listed reasons, one may wonder if the number of the deceased from the diseases in question has truly dropped or the reduction in numbers is linked to some records and the categorization of mortality into other disease categories.

Due to lack of publicly available statistics on the number of new patients and those deceased from other noncommunicable diseases for a rather long time as well as due to data inconsistency, this research has been carried out with a focus on cardiovascular diseases and diabetes. What can be emphasized as a burdening factor is the impossibility of accessing certain data owned by some institutions. In addition, life and work conditions during the COVID-19 pandemic influenced the abilities of some institutions to deliver the requested data.

A suggestion of the study is to make statistical data on noncommunicable diseases available at the level of settlements and municipalities, so that one can carry out a more-detailed geographical analysis of a disease. It is necessary for experts in various fields to take part in analyses of noncommunicable diseases, where they would examine disease causations together rather than separately. As a result, the health conditions of the population of the AP Vojvodina would be significantly improved. 

The health care institutions in the researched area should put more effort in improving the measures of psychological support offered to patients affected by noncommunicable diseases. Both medical and non-medical staff should be actively involved in providing measures of psychological support. Open communication raises hope and helps patients find strength when facing a tough situation and it simultaneously confirms that the disease has not disabled them nor excluded them from everyday life [89]. By the same token, implementing the measures of psychological support should also be directed towards patients’ families who, along with medical and non-medical staff, should take an active part in taking these measures.

By adopting a healthy lifestyle and a right approach to people’s health, it is possible to improve general health conditions and reduce the occurrence of noncommunicable diseases [90].

## 5. Conclusions

This study represents a pioneer study in Serbia which used the Getis–Ord Gi* statistic within the emerging hot spot analysis (space time pattern mining) tool and cluster analysis for the purpose of identifying “hot spots” and “colds pots”of noncommunicable diseases (cardiovascular diseases and diabetes). As presented in this research, noncommunicable diseases (cardiovascular diseases and diabetes) prevail in the pathology of diseases of the population of the AP Vojvodina. This study has cast light on the geographic medical overview of the diseases in question and identified both the most and the least burdened counties within the researched area.

By means of the obtained results of the calculated trends, it is possible to track and follow a disease trend within a specific population and to predict the further course of the disease. Consequently, the population’s health conditions can be maintained and improved. Similarly, there can be an effect on implementing the health policy measures, and it can be evaluated whether the health policy in a particular area is adequate or whether it is necessary to carry out certain modifications to the policy in accordance with the needs of the population in a certain region. By identifying the disease hot spots/cold spots and by estimating their tendencies, one can also take part in developing suitable strategies for protecting both health and the environment. Not only can research in this field contribute to the improvement of health management in a specific area but it can also influence the improvement of the overall health care system not only in the AP Vojvodina, but also in the Republic of Serbia and some other neighboring countries. 

## Figures and Tables

**Figure 1 healthcare-11-00048-f001:**
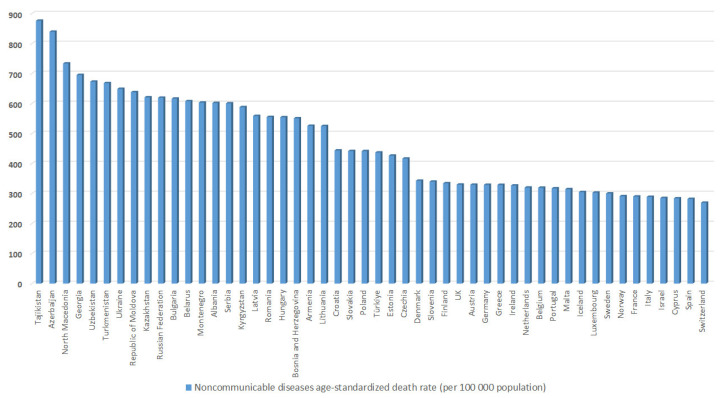
Noncommunicable diseases age-standardized death rates (per 100,000 population) in Europe in 2019. [2].

**Figure 2 healthcare-11-00048-f002:**
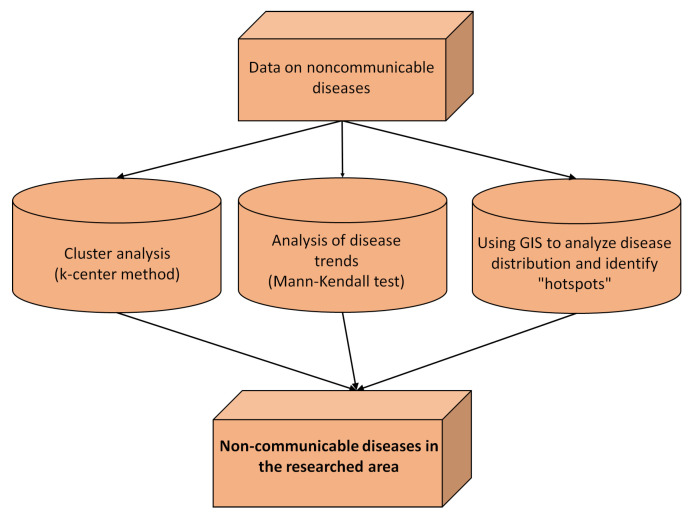
The process from the data collection to the analysis and the research results.

**Figure 3 healthcare-11-00048-f003:**
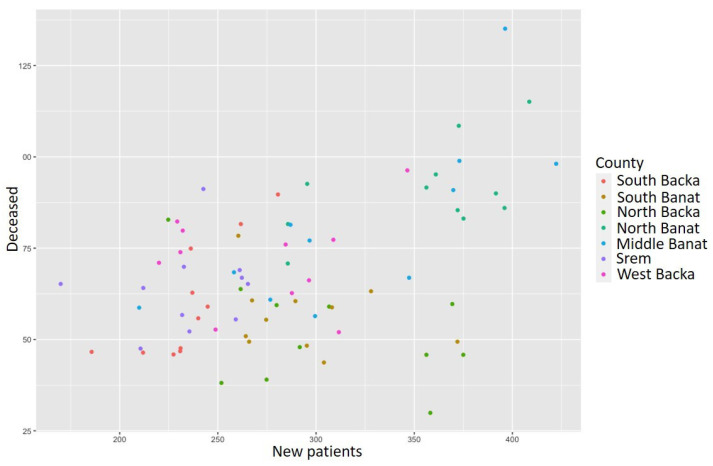
Acute coronary syndrome incidence and mortality rate ratio from 2010 to 2020.

**Figure 4 healthcare-11-00048-f004:**
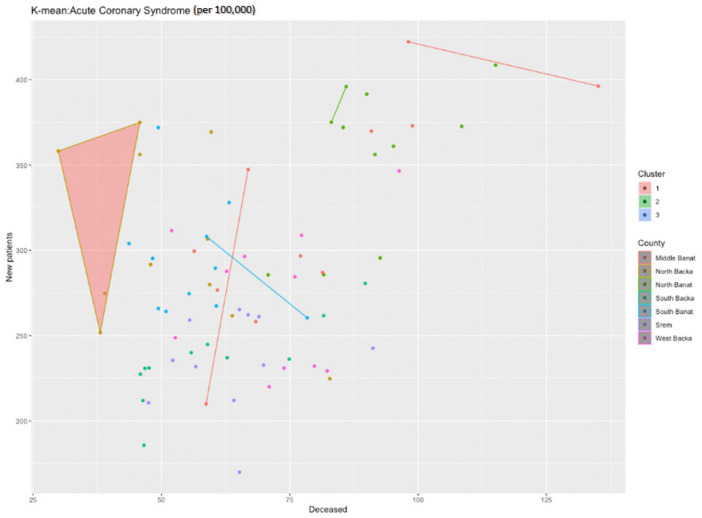
Identification of clusters during the analysis of acute coronary syndrome in the researched area.

**Figure 5 healthcare-11-00048-f005:**
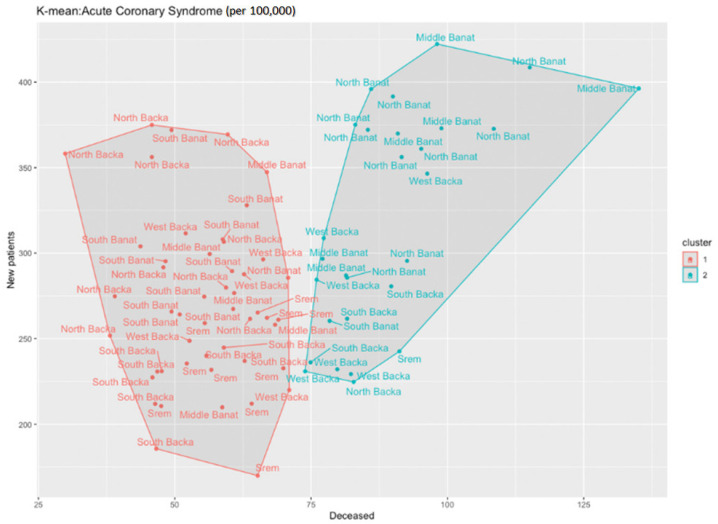
Isolation of two clusters during the analysis of acute coronary syndrome in the researched area.

**Figure 6 healthcare-11-00048-f006:**
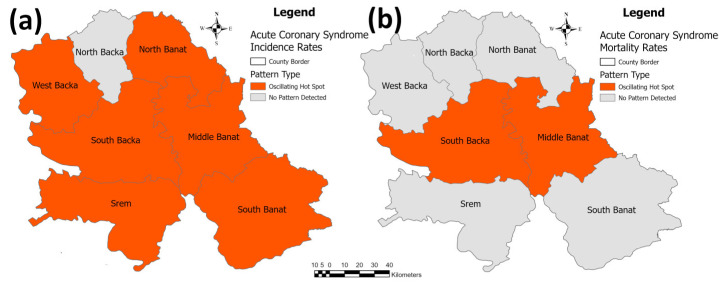
Identified patterns in the emerging hot spot analysis for the acute coronary syndrome incidence/mortality rates per 100,000 population in the province: (**a**) incidence rates, and (**b**) mortality rates.

**Figure 7 healthcare-11-00048-f007:**
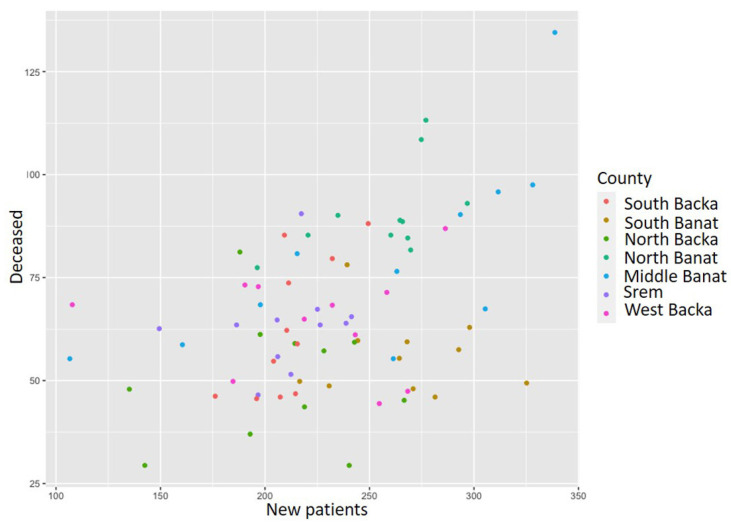
Myocardial infarction incidence and mortality rate ratio from 2010 to 2020.

**Figure 8 healthcare-11-00048-f008:**
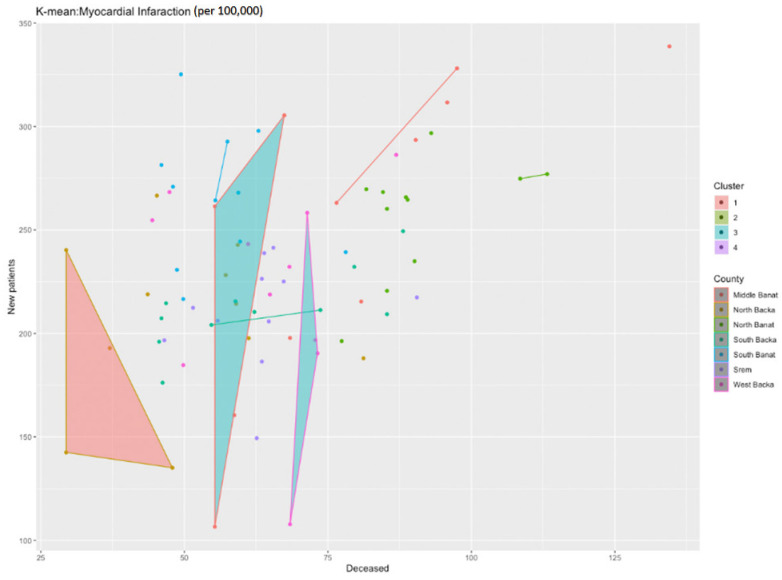
Identification of clusters in the course of analyzing myocardial infarction in the researched area.

**Figure 9 healthcare-11-00048-f009:**
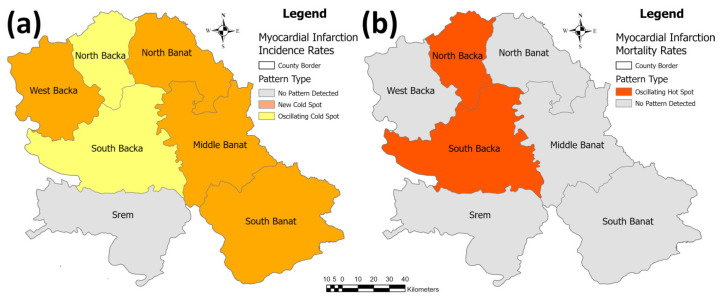
Identified patterns in the emerging hot spot analysis for the myocardial infarction incidence/mortality rates per 100,000 population in the province: (**a**) incidence rates, and (**b**) mortality rates.

**Figure 10 healthcare-11-00048-f010:**
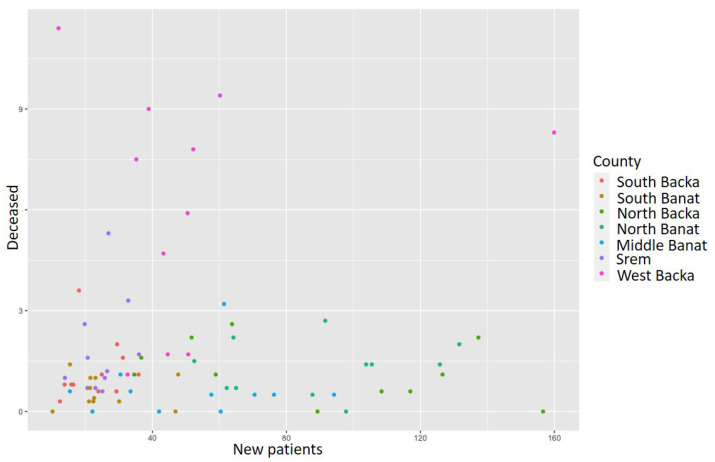
Unstable angina pectoris incidence and mortality rate ratio from 2010 to 2020.

**Figure 11 healthcare-11-00048-f011:**
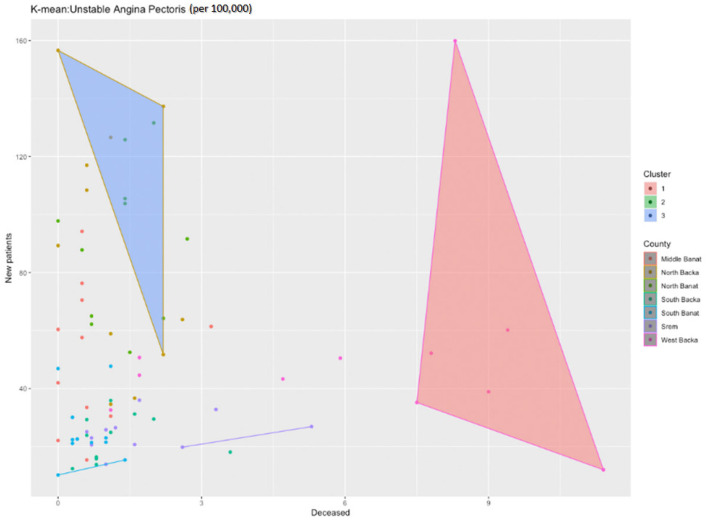
Identification of clusters during the analysis of newly diagnosed patients with unstable angina pectoris in the researched area.

**Figure 12 healthcare-11-00048-f012:**
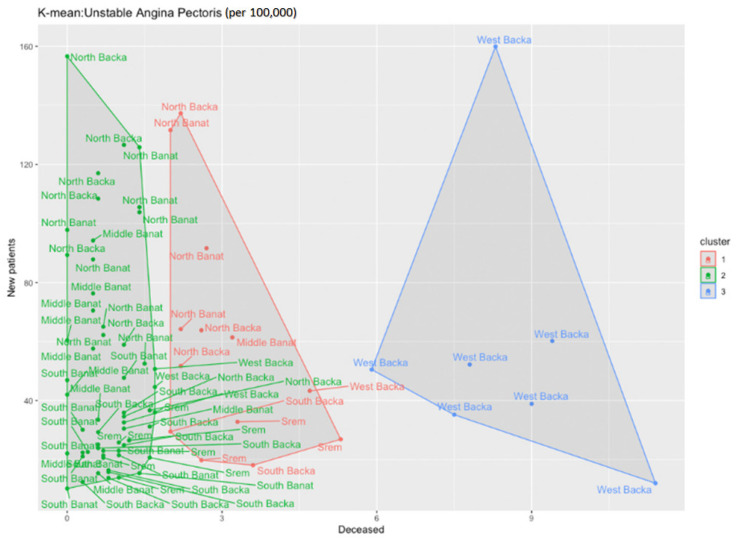
Identification of clusters during the analysis of patients deceased from unstable angina pectoris in the researched area.

**Figure 13 healthcare-11-00048-f013:**
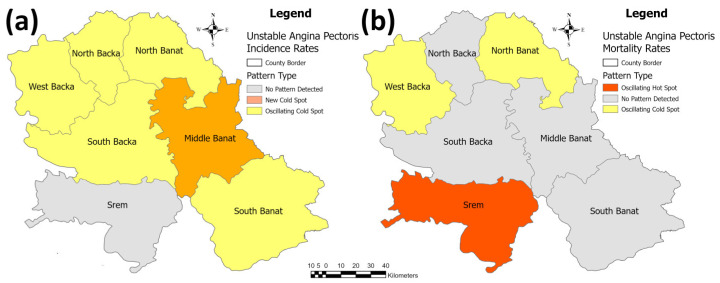
Identified patterns in the emerging hot spot analysis for the unstable angina pectoris incidence/mortality rates per 100,000 population in the province: (**a**) incidence rates, and (**b**) mortality rates.

**Figure 14 healthcare-11-00048-f014:**
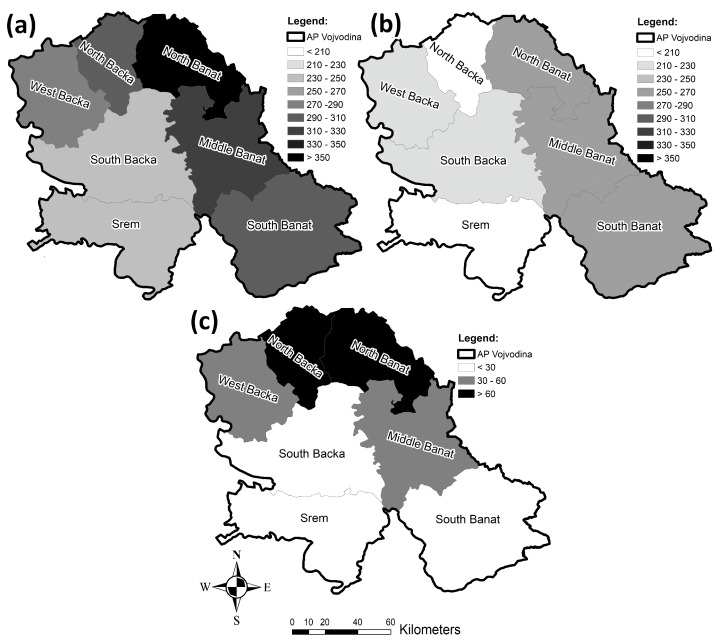
Incidence rates per 100,000 population for cardiovascular diseases in the researched area from 2010 to 2020: (**a**) acute coronary syndrome incidence rates, (**b**) myocardial infarction incidence rates, and (**c**) unstable angina pectoris incidence rates.

**Figure 15 healthcare-11-00048-f015:**
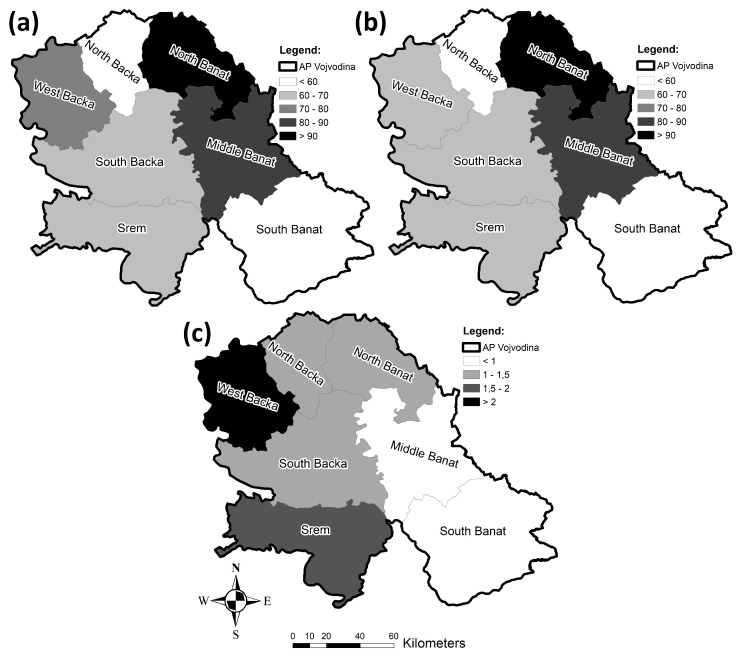
Cardiovascular diseases mortality rates per 100,000 population in the researched area from 2010 to 2020: (**a**) acute coronary syndrome, (**b**) myocardial infarction, and (**c**) unstable angina pectoris.

**Figure 16 healthcare-11-00048-f016:**
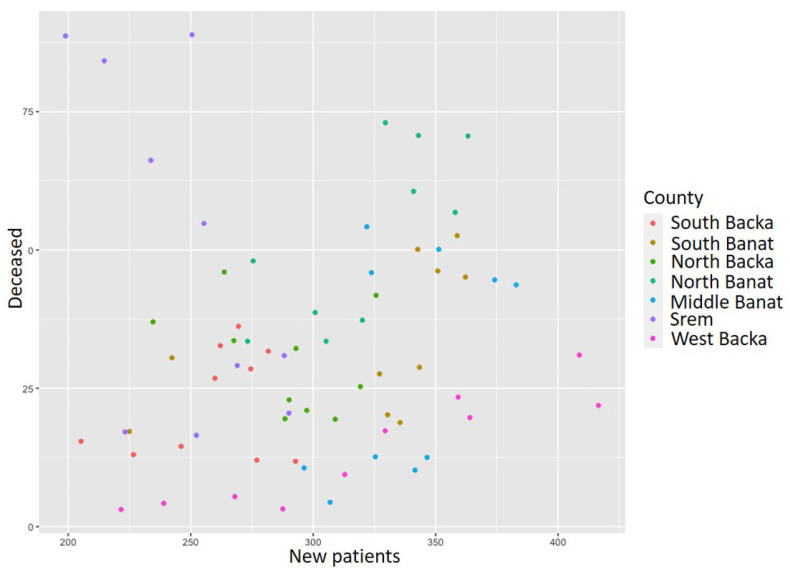
Diabetes incidence and mortality rate ratio per 100,000 population from 2010 to 2019.

**Figure 17 healthcare-11-00048-f017:**
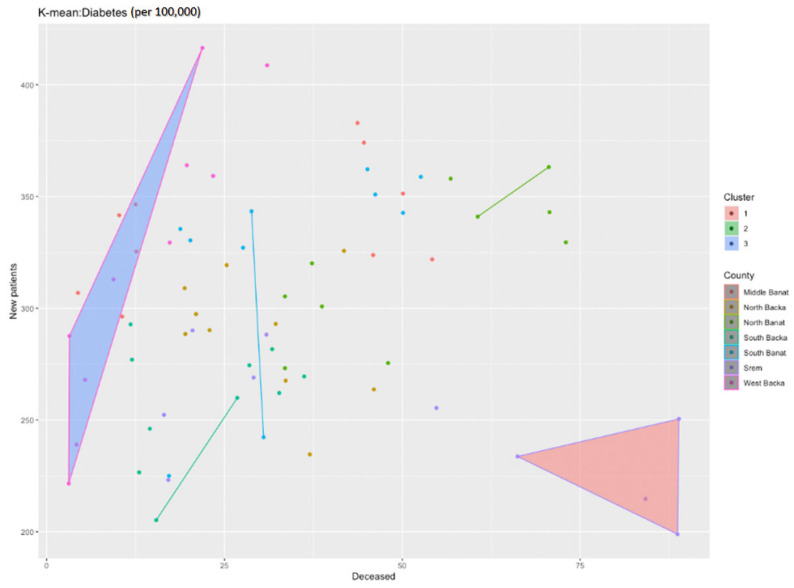
Identification of clusters during the diabetes analysis.

**Figure 18 healthcare-11-00048-f018:**
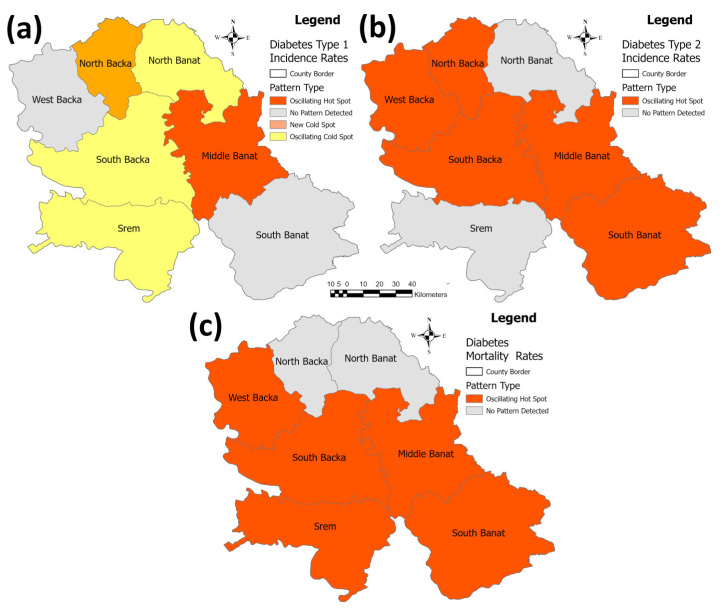
Identified patterns in the emerging hot spot analysis for the diabetes incidence/mortality rates per 100,000 population in the province: (**a**) incidence rates for type 1 diabetes (**b**) incidence rates for type 2 diabetes (**c**) mortality rates for both diabetes types.

**Figure 19 healthcare-11-00048-f019:**
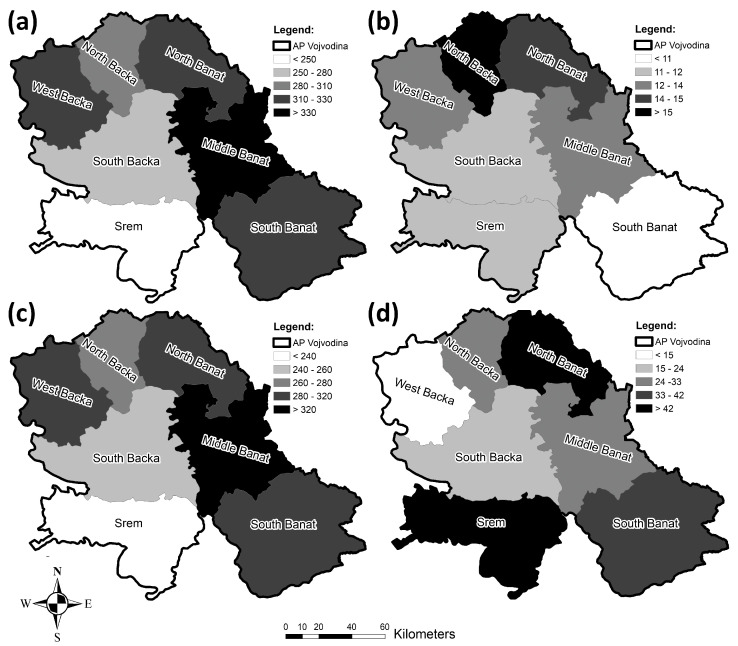
Geographical distribution of the diabetes incidence/mortality rates per 100,000 population in the researched area from 2010 to 2019: (**a**) incidence rates for both diabetes types, (**b**) incidence rates for type 1 diabetes, (**c**) incidence rates for type 2 diabetes, and (**d**) mortality rates for both diabetes types.

**Figure 20 healthcare-11-00048-f020:**
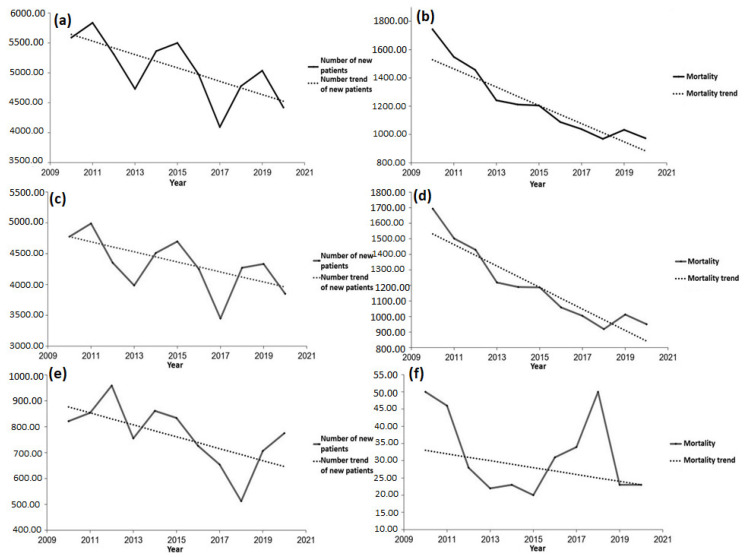
Results of the trends in the number of new patients and patients who died from cardiovascular diseases during the period 2010 to 2020: (**a**) overview of the number of new patients with acute coronary syndrome along with the trend line of the disease, (**b**) overview of the number of patients who died from acute coronary syndrome with the mortality trend, (**c**) overview of the number of new patients with myocardial infarction along with the trend in the number of new patients, (**d**) overview of the number of patients who died from myocardial infarction and the mortality trend, (**e**) overview of the number of new patients with unstable angina pectoris with the trend line of new patients, and (**f**) overview of the number of patients who died from unstable angina pectoris with the mortality trend line.

**Figure 21 healthcare-11-00048-f021:**
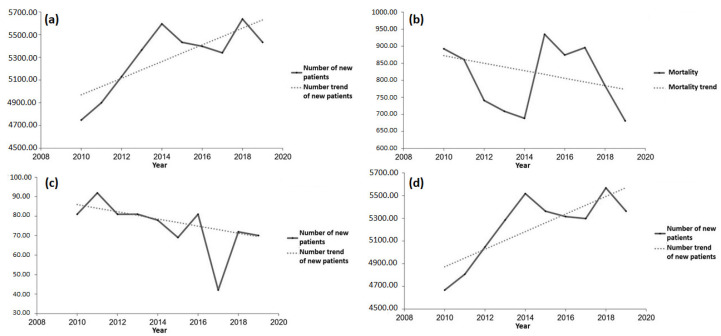
Results of the trends showing the number of new patients and those deceased from diabetes in the researched area: (**a**) overview of the number of new patients with both diabetes types from 2010 to 2019, (**b**) overview of the number of people who died from both diabetes types from 2010 to 2019, (**c**) overview of the number of new patients with type 1 diabetes from 2010 to 2019, (**d**) and overview of the number of new patients with type 2 diabetes from 2010 to 2019.

**Table 1 healthcare-11-00048-t001:** Parameters of the Mann–-Kendall test for particular diseases in the researched area (α—significance level, +α = 0.1, * α = 0.05, *** α = 0.001, /—no significance; *b*—so called Sen’s slope).

Type of Disease	Number of New Patients			Number of the Deceased		
	*Z*	*α*	*b*	*Z*	*A*	*b*
Acute coronary syndrome	−2.02	*	−0.0112	−3.89	***	−0.643
Myocardial infarction	−1.87	+	−0.816	−3.74	***	−0.690
Unstable angina pectoris	−1.87	+	−0.230	−0.63	/	−1.000
Type 1 diabetes	−2.23	*	−1.833	−0.72	/	−0.110
Type 2 diabetes	2.42	*	0.773			

## Data Availability

To obtain the data for this study, please contact the authors via email.

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
