# Peer review of "Geographic Medical Overview of Noncommunicable Diseases (Cardiovascular Diseases and Diabetes) in the Territory of the AP Vojvodina (Northern Serbia)"

_healthcare, 2022, doi:10.3390/healthcare11010048_

Round 1

Reviewer 1 Report

In my opinion, there are many issues to solve here:

1 - The introduction chapter is somewhat dispersed, in the sense that it seems to focus worldwide and in many specific geographic areas other than in the research area, for instance. Also, the discussion of study objectives at the end of the introduction is rather poor and it does not bring any new information.

2 – The map at figure 2 is a bit confusing, it is not clear where exactly is the research area, since more than an area is highlighted and there is no clear legend relating to the text (I had to consult google maps eventually).    

3 – The English needs to be strongly revised and there are many typing errors.

4 – The term “hotspots” is used in a rather confusing manner, since in ESRI software this usually means using the Hot Spot Analysis (Getis-Ord Gi* spatial clustering statistic method) but I don´t see it referred or explained anywhere in the text. Although it always represents a starting point in exploratory spatial data analysis, the use of cartography/map visualization is not very efficient to detect hotspots/spatial clusters, since it does not offer statistical significance evaluation of clustering and clusters.    

5 – Expanding on the previous point, since this is/should be mainly a process of spatial analysis, the use of a non-spatial clustering technique (although possible) makes it poorer and more difficult in terms of interpretation. Spatial cluster analysis methods (such as Getis-Ord Gi*, Local Moran, etc), should have been at least used and compared with its non-spatial counterpart, to improve the analysis.  

Author Response

Dear,

Thank you very much for your reviews. We revised the manuscript according to your review as follows:

  1. The introduction is revised and the reports from outside of Europe are excluded from the manuscript (according to another review). A study from the researched area regarding diabetes disease is cited. Also, we added up-to-date data in regard to the current situation with NCDs (diabetes and CVD).
  2. According to Editor’s Notes, we excluded the map in Figure 2.
  3. The English language is revised by an English language specialist.
  4. We added the Emerging Hot Spot Analysis (Space Time Pattern Mining) tool from ArcGIS which uses Getis-Ord Gi* spatial clustering statistic method and this method is added to the manuscript methodology for each disease.
  5. New maps in Figures 6, 9, 13 and 18 are added and the results from this spatial analysis are compared to the non-spatial counterpart.

Kind regards.

Reviewer 2 Report

A brief summary

The aim of the study is to identify counties with most burden in cardiovascular diseases and diabetes. The authors use cluster analysis and Mann-Kendall test. It seems that official statistics are not available easily regarding the matter. The authors use different data sources and provide a good overview of the incidence and mortality rates in the study region.

General comments:

The authors use different data sources and provide a good overview of the incidence and mortality rates in the study region. The analysis is done on the county and province level. The authors provide a map with the study region (counties) in page 14. It would be easier for the reader to follow up the manuscript if the study region map would be presented earlier. The analysis methods used are beyond my expertise so I cannot comment on the methods and its interpretations (cluster analysis, Mann-Kendall test). However, I think that in the discussion section it should be discussed whether the methods were useful and could the methods be applied for similar settings elsewhere.

Specific comments:
- Page 2. Is it necessary to report deaths etc. for different countries (e.g. India and Tanzania)? Maybe overall global situation and the situation in Europe would be enough.
- Page 2. I would prefer better and up to date references for the global diabetes and cardiovascular diseases situation than Ministry of Health of the Republic of Serbia from 2013 and WHO from 2013. I recommend checking out WHO for diabetes and for cardiovascular diseases and IDF Diabetes Atlas, for example.
- Page 3. Figure 1: The figure presents the number of premature deaths. It would be easier to compare the countries if it would be per 100.000 inhabitants, for example.
- Page 4. What do you mean by “boasts a good geographical position”?
- Figure 2: Could you please provide a legend for the map. In addition, it would be good to have a smaller map with the study region counties.
- Page 6. The authors write “it is important to analyse the distribution of the diseases from biomedical, economic and behavioural perspectives”. Is it necessary to mention these because these are not analyzed in the article?
- Page 7. What the authors mean by this: “a significant place is occupied by blood circulatory system diseases and diabetes”?
- Page 7. Figure 4. Please explain what is seen in the figure 4. What is the purpose of figure 4? Are the dots for different years? Is the figure necessary? Newly diagnosed and deceased cases are anyway present in tables A1, A2 and A3.
- Page 8. Figure 5. What do the numbers in the figure stand for?
- Page 10. Figure 7. Is the figure necessary? Newly diagnosed and deceased cases are anyway present in tables A1, A2 and A3.
- Page 12. Figure 9. Is the figure necessary? Newly diagnosed and deceased cases are anyway present in tables A1, A2 and A3.
- Page 14. Figure 12. Please provide the unit for the incidence rate. Is it per 100.000 people? Map A and B could be compared if the groups in the map would be classified the same way (same number of classes and thresholds). Now there are 4 classes in A with >350 the highest category and 6 classes in B with >260 the highest category.
- Page 15. Figure 13. Same comment as for figure 12.
- Page 16. Figure 15. Is the figure necessary? Newly diagnosed and deceased cases are anyway present in tables A1, A2 and A3.
- Page 21. The authors discuss about the effects that are associated with the studied diseases. I think ageing of the population should be mentioned. The age structure of the counties might influence the incidence and mortality rates. What about, how is the public health care organized in the country?
- Page 22. The authors write that “By reducing the environment pollution, we can reduce, to a great extent, frequency of the noncommunicable diseases occurrence in the research area.” I think this is quite strong statement especially because the association of environmental pollution and cardiovascular diseases are not studied in the study.
- I recommend language editing.

Author Response

Dear,

Thank you very much for your reviews. We revised the manuscript according to your review as follows:

  1. Page 2. Is it necessary to report deaths etc….: The reports outside of Europe are excluded from the manuscript and data on Europe’s situation regarding NCDs and diabetes separately are given using IDF Diabetes Atlas (data from 2021) and WHO’s NCD Portal (data from 2019).
  2. Page 2. I would prefer better and up to date references…: The Ministry of Health of the Republic of Serbia didn’t issue any other recent national guide regarding good clinical practice in diabetes mellitus, and that is the reason why we used this from 2013. The WHO didn’t issue any other recent Global action plan, apart Global action plan for the prevention and control of noncommunicable diseases (NCDs) 2013-2020, and we considered it valuable to use it, due to the research period from 2010 to 2020. Now, regarding the current situation considering NCDs we added data from IDF Diabetes Atlas, 10th edition (2021) and WHO’s NCD Portal (data from 2019).
  3. Page 3. Figure 1…: The Chart with the numbers of premature deaths is changed with the chart of Europe’s NCD mortality rate from WHO’s NCD Portal (data from 2019).
  4. Page 4. What do you mean by…: This is cited from the book in Serbian language and now we used another word, but it has the same meaning – this province has a favourable geographical position.
  5. Figure 2: Could you please provide…: This figure is excluded according to the editor’s suggestion.
  6. Page 6. The authors write “it is important…: we excluded this from the manuscript because it would demand a great time consumption if we tried to analyse it in the article.
  7. Page 7. What the authors mean by…: we revised the sentence and believe that now is more clear that the blood circulatory system diseases and diabetes are the most significant health problem of the population in the research area. All the publications that are mentioned in this sentence state this. It is usually at the beginning of these reports. For example, in [49.] on Page 2, the first paragraph explains this statement.
  8. Page 7. Figure 4. Please explain what is seen ….: We added an explanation and the purpose of the figure. Yes, data are available in tables, but we wanted to check if there is simple linear regression in the data in order to use the appropriate statistic for analysis.
  9. Page 8. Figure 5. What do the numbers in the figure stand for? The numbers in the figure represent the county serial number in the database. We changed the image in the figure and excluded numbers from the image.
  10. Page 10. Figure 7. Is the…: We added an explanation and the purpose of the figure. Yes, data are available in tables, but we wanted to check if there is simple linear regression in the data in order to use the appropriate statistic for analysis. (Same as for Figure 4.)
  11. Page 12. Figure 9. Is the figure necessary? …: We added an explanation and the purpose of the figure. Yes, data are available in tables, but we wanted to check if there is simple linear regression in the data in order to use the appropriate statistic for analysis. (Same as for Figure 4. and Figure 7.)
  12. Page 14. Figure 12. Please provide…: We added a unit for the incidence rate. Also, we revised maps A and B in the Figure and now the thresholds and number of classes are the same.
  13. Page 15. Figure 13. Same comment as for figure 12…: We revised maps A and B in the Figure and now thresholds and number of classes are the same.
  14. Page 16. Figure 15. Is the figure necessary?...: We added an explanation and the purpose of figure 14. Yes, data are available in tables, but we wanted to check if there is simple linear regression in the data in order to use the appropriate statistic for analysis. (Same as for Figure 4, Figure 7. and Figure 9.) Figure 15 represents the identification of clusters and we believe this chart is valuable for the analysis in the manuscript. The explanation regarding this Figure is already in the manuscript.
  15. Page 21. The authors discuss about the …: We added the age structure discussion in the manuscript. Data regarding national and regional levels are cited from the official Serbian statistic office and for Europe is cited from the EUROSTAT website. We added the healthcare system organisation in Serbia and the AP Vojvodina in the introduction part of the manuscript.
  16. Page 22. The authors write that “By reducing…: We excluded this sentence from the manuscript as well as the following one.
  17. I recommend language editing…: The English language is revised by an English language specialist.

We also added the Emerging Hot Spot Analysis (Space Time Pattern Mining) tool from ArcGIS that uses the Gettys-Ord Gi* statistical spatial clustering method as suggested by another reviewer.

Round 2

Reviewer 1 Report

I suggest, in line 17,  that  "Getis-Ord Gi* statistical software for the hot spot analysis..."   be changed to  "Getis-Ord Gi* method for hot spot analysis..."